# Temporal Variation in the Genetic Composition of an Endangered Marsupial Reflects Reintroduction History

**Rujiporn Thavornkanlapachai** [1,*], **Harriet R. Mills** [2], **Kym Ottewell** [3], **J. Anthony Friend** [4] **and W. Jason Kennington** [1]

1   School of Biological Sciences, The University of Western Australia, Crawley, WA 6009, Australia; jason.kennington@uwa.edu.au
2   Centre for Ecosystem Management, School of Science, Edith Cowan University, Joondalup, WA 6027, Australia; harriet.mills@ecu.edu.au
3   Department of Biodiversity, Conservation and Attractions, Locked Bag 104, Bentley Delivery Centre, Bentley, WA 6152, Australia; kym.ottewell@dbca.wa.gov.au
4   Department of Biodiversity, Conservation and Attractions, Albany Highway 120, Albany, WA 6330, Australia; tony.friend@dbca.wa.gov.au
*   Correspondence: rujiporn.sun@dbca.wa.gov.au; Tel.: +61-8-9219-9089

**Abstract:** The loss of genetic variation and genetic divergence from source populations are common problems for reintroductions that use captive animals or a small number of founders to establish a new population. This study evaluated the genetic changes occurring in a captive and a reintroduced population of the dibbler (*Parantechinus apicalis*) that were established from multiple source populations over a twelve-year period, using 21 microsatellite loci. While the levels of genetic variation within the captive and reintroduced populations were relatively stable, and did not differ significantly from the source populations, their effective population size reduced 10–16-fold over the duration of this study. Evidence of some loss of genetic variation in the reintroduced population coincided with genetic bottlenecks that occurred after the population had become established. Detectable changes in the genetic composition of both captive and reintroduced populations were associated with the origins of the individuals introduced to the population. We show that interbreeding between individuals from different source populations lowered the genetic relatedness among the offspring, but this was short-lived. Our study highlights the importance of sourcing founders from multiple locations in conservation breeding programs to avoid inbreeding and maximize allelic diversity. The manipulation of genetic composition in a captive or reintroduced population is possible with careful management of the origins and timings of founder releases.

**Keywords:** dasyurid; genetic mixing; subpopulation; multiple reintroduction; relatedness; microsatellite

## 1. Introduction

Many species have experienced declines in their abundance and distribution, or have become extinct as a result of human activities [1]. As these threats continue to endanger native populations, translocation, a conservation tool that involves moving individuals from one location to another, is frequently implemented to recover population numbers. There are different types of translocations with specific aims as followed: to restore existing (reinforcement) and locally extirpated (reintroduction) populations; and to introduce individuals outside their natural distributional range because their historical range is no longer suitable (assisted colonization), or to perform a specific ecological function (ecological replacement) [2]. The success of a translocation is influenced by various factors, including the efficiency in the removal of the threat(s), the habitat quality, size of released area, and the number of individuals released [3–6]. Recently, genetic approaches have become important in evaluating appropriate source populations for release as well as for ongoing monitoring to assess whether there has been loss of genetic diversity or inbreeding [7–9].

The translocated and captive populations are prone to a loss of genetic diversity and inbreeding due to the founder effects, that is, establishing a new population with a limited number of individuals (e.g., [10,11]). Small numbers of founders often result in small effective population sizes, leading to fluctuations in the allele frequencies and genetic divergence between the new population and its source via genetic drift [12–14]. In addition, establishing new populations using individuals selected from inbred wild populations (e.g., [15–17]) or captive populations (e.g., [18,19]) has the potential to further increase inbreeding and reduce fitness (e.g., [20]). The reduced fitness of individuals in translocated populations may ultimately lead to a failure of that translocation if no conservation intervention is undertaken.

Translocations can be managed in ways to bolster genetic diversity in the translocated population and minimize further losses. First, sourcing individuals from multiple populations can maximize adaptive potential and may confer fitness benefits. Many translocations sourcing from multiple populations have shown an increased genetic diversity and reduced inbreeding over multiple generations [9,21–23]. However, the initial genetic contribution of founders from different sources can be affected by uneven mortality, release time, proportion of different founder sources, and variance in reproductive success [23–26]. For example, the initial reintroduction of the burrowing bettong (*Bettongia lesueur*) back to mainland Australia showed to have ancestral genetic proportions reflecting the proportion of founder sources after the known mortality was removed [23]. The genetic admixture was also delayed by a different release time of the second founder source and deviated from the initial phase over time as a result of the poor recruitment of one of the source populations [26]. Second, the number of released animals should be adequately large enough to capture at least 95% of heterozygosity or rare alleles of frequency <5% of the source population(s) [27,28]. Captive breeding is often needed to achieve this number due to many threatened species having a small source population size and/or to avoid impacts on the remaining population(s). To maintain genetic diversity in a new population, multiple releases have shown to replenish the initial losses of individuals during the establishment phase (e.g., [29]).

The dibbler (*Parantechinus apicalis*) is a small (40–125 g) insectivorous marsupial [30–32] endemic to Western Australia (WA). *Parantechinus apicalis* were once widely distributed in WA, from Shark Bay on the central western coast to Esperance on the southern coastline, and east to the Eyre Peninsula, South Australia [33–35]. Their current distribution is restricted to two small islands, Boullanger and Whitlock Islands, off the coast from Jurien Bay on the western coastline and in the Fitzgerald River National Park (FRNP: ~3300 km$^2$) on the mainland, 200 km west of Esperance (Figure 1; [36–38]). They are seasonal breeders, breeding once a year around March to April [39]. A female produces up to eight young per breeding season [40], with the young reaching sexual maturity after 10 to 11 months [32]. On the mainland, the female dibblers can live up to four years and males up to three years [41]. While Boullanger Island dibblers exhibit facultative male die-off after the first breeding season in some years, the mainland male dibblers have been reported to survive well into their second year [41].

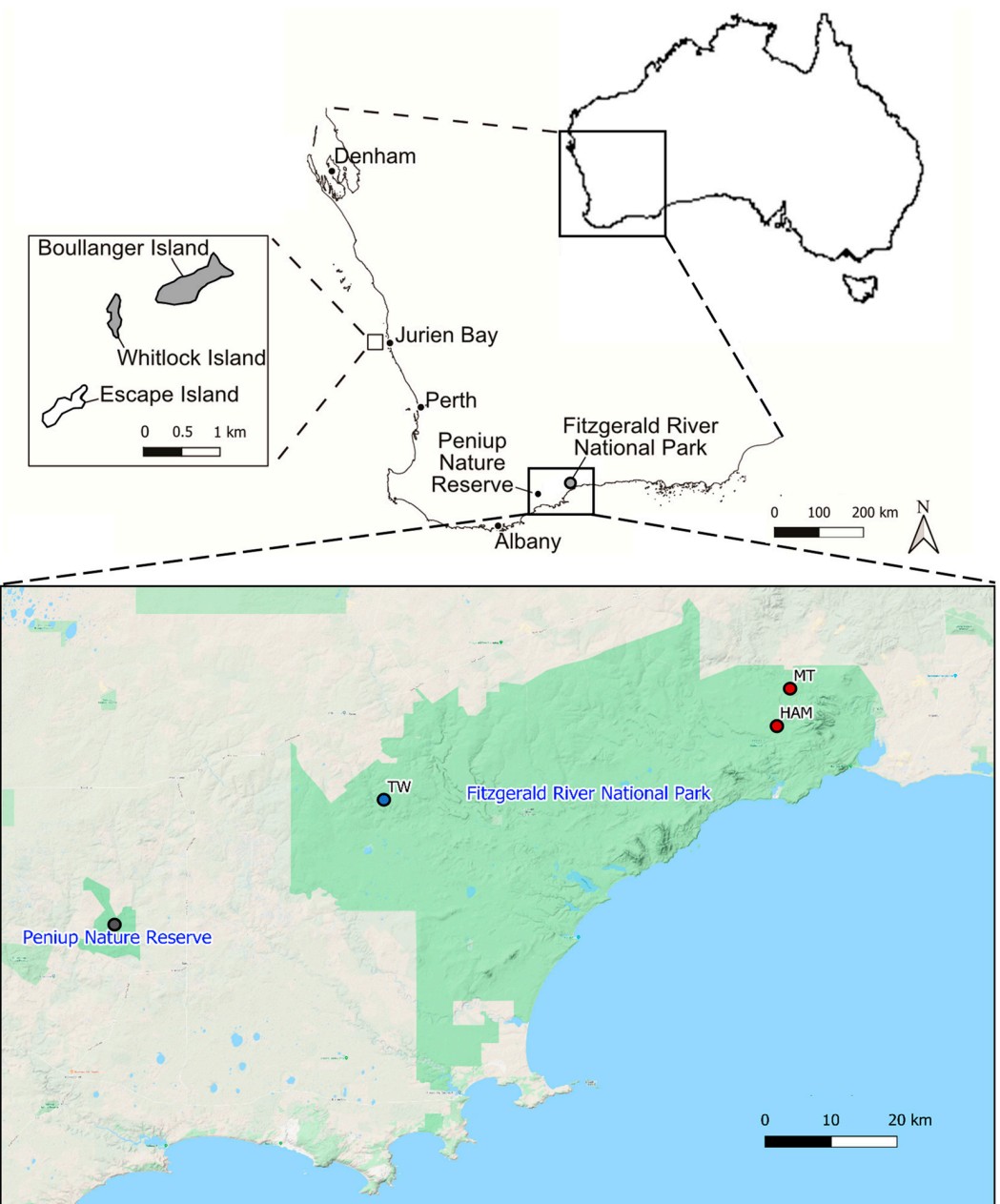

**Figure 1.** Map of *Parantechinus apicalis* trapping sites within the Fitzgerald River National Park and the location of the reintroduced population at Peniup Nature Reserve (adapted from Aisya et al. [42] under review). Hamersley Moir (HAM) and Moir Track (MT) represent the eastern subpopulation and Twertup (TW) the western subpopulation. The top inset shows locations of the remaining wild populations (light grey shaded).

Dibblers are currently listed as endangered under the Environment Protection and Biodiversity Conservation Act (1999) and the 2016 IUCN Red List of Threatened Species [43]. The main threats to dibblers include introduced predators such as foxes and feral cats, inappropriate fire regimes, habitat degradation and, on Boullanger and Whitlock Islands, competition with the introduced house mouse, *Mus musculus* [33]. These threats have resulted in a steady decline in the dibbler population sizes. There are currently less than 1000 mature individuals estimated to be on the mainland [44] and approximately 100 individuals known to be alive on both islands [29]. In a bid to bolster the declining numbers, a captive breeding program was established in 1997 at Perth Zoo, using island animals to generate stock for introductions to Escape Island [29]. In 2000, the breeding

program was converted to mainland stock, using wild dibblers from the FRNP population. In 2001, the captive born dibblers were released to Peniup Nature Reserve (~30 km west of the FRNP). A further six releases were conducted from the captive population to the reintroduced population over the next nine years.

Here, we evaluate the success of the Peniup Nature Reserve reintroduction using a longitudinal genetic dataset to retrospectively assess genetic diversity and inbreeding in the reintroduced population. A previous study on the mainland population confirmed two genetically distinct sub-populations on the western and eastern sides of the FRNP, respectively [45]. Both populations were used in the captive breeding program, but it is unknown if both sub-populations were successfully established. The objective of this study is to determine the relative success of the Peniup reintroduction in maintaining genetic variation relative to the wild source populations, and to examine the extent of admixture within captive and reintroduced populations over a twelve-year period.

## 2. Materials and Methods

### 2.1. Sample Collection

The Perth Zoo captive dibbler population was established using 26 individuals, referred to as founders hereafter, collected over multiple years from several sites in the Fitzgerald River National Park, Western Australia (33°52′ S, 119°54′ E) (Figure 1, Table 1). Pairings between individuals in captivity were determined using a minimum kinship design whereby each sex was ordered according to their minimum kinship estimates, and the males and females with the lowest estimate were paired together, and so on until all adults had been allocated a partner. From this captive population, 218 captive-born dibblers as well as 17 of the original founders were released to the reintroduction site at Peniup Nature Reserve (34°10′ S, 118°49′ E) in October 2001–2003, 2006, and 2008–2010 (Table 2). A total of 133 samples were collected from wild-born animals at the reintroduction site during follow-up monitoring 2002–2012 (Figure 1, Table 1). In addition to samples from the captive and reintroduced populations, samples were collected from each of the source populations during their regular monitoring between 2000 and 2012. A total of 156 samples were collected from the eastern source population at Hamersley Moir (HAM) and Moir Track (MT) (33°53′ S, 119°55′ E) and 49 samples from the western source population at Twertup (TW) (33°58′ S, 119°16′ E). All sampled individuals had a biopsy punch (~1 mm$^2$) taken from their ear, a microchip implanted, and their sex recorded. Ear tissue samples were stored in 20% dimethyl sulfoxide (DMSO) saturated with sodium chlorine (NaCl) at room temperature. All sample collections were under animal ethic approvals by the University of Western Australia Animal ethics committee (AEC: 16A/2012), the Zoological Parks Authority Animal ethics committee (SOP 49,063 and 24252), and the Department of Biodiversity, Conservation and Attractions (DEC AEC: 66/2009).

**Table 1.** Summary of *Parantechinus apicalis* samples used in this study. Eastern and western sources represent the following source populations and locations where wild-born animals were trapped: Hamersley Moir (HAM), Moir Track (MT) and Twertup (TW). Founders are individuals selected from the source populations to breed in the captive colony at Perth Zoo. 'Captive' represents animals born in captivity between 2000 and 2010. 'Released' represents animals that were released to the Peniup Nature Reserve between 2001 and 2010. This includes both captive and wild-born animals. Peniup represents wild-born animals caught at the reintroduction site.

| | Source Populations | | | Founders | | | | |
|---|---|---|---|---|---|---|---|---|
| **Year** | **West** | **East** | | **West** | **East** | **Captive** | **Released** | **Peniup** |
| | **TW** | **HAM** | **MT** | | | | | |
| 2000 | 11 | | | 7 | | 8 | | |
| 2001 | 2 | | | 1 | 3 | 41 | 41 | |
| 2002 | 3 | | | | | 39 | 46 | 5 |
| 2003 | 4 | | 3 | | | 36 | 43 | 14 |

**Table 1.** *Cont.*

| Year | Source Populations | | | Founders | | Captive | Released | Peniup |
|---|---|---|---|---|---|---|---|---|
| | West | East | | West | East | | | |
| | TW | HAM | MT | | | | | |
| 2004 | 18 | | 3 | 6 | 3 | | | 43 |
| 2005 | 10 | 45 | 5 | | | 3 | | |
| 2006 | | 17 | | | | 15 | 6 | 11 |
| 2007 | | 22 | | | 7 | 5 | | 3 |
| 2008 | | 18 | | | | 19 | 24 | |
| 2009 | | 5 | | | 3 | 20 | 34 | 7 |
| 2010 | | 17 | | | | 37 | 41 | 14 |
| 2011 | 1 | 2 | | | | | | 21 |
| 2012 | | 19 | | | | | | 15 |

**Table 2.** Summary of released *Parantechinus apicalis* to Peniup Nature Reserve between 2001 and 2010. Brackets indicate numbers of founders that were released after contributing offspring to the captive program.

| Year of Release | Sex | Age (Year) | | | | Total |
|---|---|---|---|---|---|---|
| | | <1 | 1 | 2 | 3 | |
| 2001 | Female | 16 | 3 | [1] | [1] | 21 |
| | Male | 14 | 4 | [2] | | 20 |
| 2002 | Female | 18 | 2 | [3] | | 23 |
| | Male | 16 | 5 | [2] | | 23 |
| 2003 | Female | 16 | 1 | 3 | | 20 |
| | Male | 20 | 2 | 1 | | 23 |
| 2006 | Female | 4 | | | | 4 |
| | Male | 2 | | | | 2 |
| 2008 | Female | 5 | | 1 [1] | 2 [1] | 10 |
| | Male | 8 | | 3 [2] | 1 | 14 |
| 2009 | Female | 4 | 2 | [1] | 3 [1] | 11 |
| | Male | 14 | 3 | 2 [1] | 2 [1] | 23 |
| 2010 | Female | 22 | | 1 | 1 | 24 |
| | Male | 15 | 1 | | 1 | 17 |
| Total | | 174 | 23 | 24 | 14 | 235 |

DNA was extracted using the 'salting-out' method [46] with a modification of a 56 °C incubation step and 10 mg/mL of proteinase K being added to 300 µL TNES. Each individual was genotyped using the following 21 microsatellite loci developed for *P. apicalis* and other dasyurids: pPa2D4, pPa2A12, pPa2B10, pPa7A1, pPa7H9, pPa9D2, pPa1B10, pPa4B3, pPa8F10 (*P. apicalis*, [47]); pDG1A1, pDG1H3, pDG6D5 (*Dasyurus geoffroii*, [48]; 3.1.2, 3.3.1, 3.3.2, 4.4.2, 4.4.10 (*Dasyurus spp.*, [49]); Sh3o, Sh6e (*Sarcophilus laniarius*, [50]); Aa4A (*Antechinus agilis*, [51]), Aa4J (*A. agilis*, [52]). PCRs (volume 10 µL) were performed using a QIAGEN multiplex PCR kit and contained primer concentrations ranging from 0.04 to 1.5 µm and 10–20 ng of DNA (Table S1). Amplifications were performed using an Eppendorf mastercycler epgradientS thermocycler with the following steps: 15 min at 95 °C, 35 to 40 cycles at 94 °C for of 30 s, the annealing temperature (46 °C to 58 °C) for 90 s, 72 °C for 60 s, and finally 60 °C for 30 min (Table S1). PCR products were analyzed in an ABI 3730 sequencer using a GeneScan-600 LIZ internal size standard and scored using GeneMarker version 1.90 (SoftGenetics).

### 2.2. Data Analysis

Genotype quality was assessed by calculating the allele-specific and locus-specific genotypic error rates [53]. We tested for the presence of null alleles in the source population samples at each locus using Microchecker [54]. We analyzed samples from each population

by collection year when $N \geq 10$ and as pooled samples (all collection years analyzed together). Microsatellite variation was quantified by calculating the allelic richness ($A_r$) (the allele number per locus estimate corrected for sample size) and gene diversity ($H$). Deviations from Hardy–Weinberg equilibrium were assessed by calculating the inbreeding coefficient ($F_{IS}$) and randomization tests were performed to test the significance of the deviations. Positive $F_{IS}$ values indicate a deficit of heterozygotes, while negative $F_{IS}$ values indicate an excess of heterozygotes. Randomization tests were also performed to test for genotypic disequilibrium between each pair of loci. For these tests, the sequential Bonferroni correction [55] was applied to control for type I statistical error. Genetic differentiation between population samples were quantified using Weir & Cockerham's [56] $F_{ST}$ and were assessed for significance using randomization tests. All above genetic parameters and tests were calculated using FSTAT version 2.9.3.2 [57]. The number of rare alleles ($A_{(rare)}$) with frequency less than 5% and the number of unique alleles ($A_u$) were calculated in GENALEX version 6.5 [58]. Differences in $H$, $A_{(rare)}$, $A_u$, and $A_r$ between collection years and pooled sample populations were tested using Wilcoxon's signed-rank tests with loci as the pairing factor using the R version 3.5.1 statistical package [59].

The effective population size ($N_e$) for each population sample and samples pooled across collection years were estimated using the single-sampled estimator of $N_e$ as implemented in the software package LDN$_E$ [60]. We assumed that all of our population samples consisted of overlapping generations. We used a random mating model and estimated linkage disequilibrium amongst alleles using only alleles with frequencies >5%, as this was expected to give the best balance between precision and bias in the $N_e$ estimator [61].

The occurrence of recent reductions in $N_e$ was investigated by testing for an excess in heterozygosity using the program Bottleneck [62]. Both the stepwise mutation model (SMM) and two-phase model (TPM) were used. These models were chosen because they are considered to be the most appropriate for microsatellite data [62]. Variance for TPM was set to 12 and the proportion of SMM in TPM was 95% with 1000 iterations following approaches described by Luitkart and Cornuet [63] and Luikart et al. [64].

To investigate the extent of genetic mixing between the eastern and western source population subpopulations within the captive and reintroduced populations, we used a discriminant analysis of principal components (DAPC) provided in the Adegenet package version 2.0.1 [65,66] in the R version 3.5.1 statistical package [59]. DAPC grouped individuals to achieve the largest between-group variance and the smallest within-group variance using linear combinations of alleles [66]. To achieve this, principal component analysis is performed as a prior step to the discriminant analysis. We ran the *find.cluster* command with the number of components (PCs) that allowed 90% of cumulative variance to be retained (between 40–50 PCs) and selected two clusters based on the number of source populations. Then we ran the *dapc* command on samples using sampling locations or collection years as their assigned groups. We retained the number of PCs as indicated by *find.cluster* command and the number of the discriminant functions as the number of groups-1.

Finally, pairwise relatedness estimates were calculated using the method of Lynch and Ritland [67] implemented in GENALEX version 6.5 [58]. Differences in pairwise relatedness between population samples were tested using Wilcoxon's signed-rank tests implemented in the statistical package R version 3.5.1 [59]. Confidence limits for population mean values were calculated using bootstrapping (1000 bootstraps) in R.

## 3. Results

### 3.1. Effects of Reintroduction on Genetic Variability

The allele-specific and locus-specific genotyping error rates were 0.016 and 0.026, respectively. The average amplification success rate per locus was 0.946. Microchecker identified one locus (aPa1B10) as having null alleles in both of the source populations. This locus was removed from further analysis.

Overall, the estimates of genetic diversity of the captive and reintroduced populations were lower than the source populations (Figure 2a,b, Table 3). This pattern was consistent over multiple years. The population samples from the reintroduced population in the years 2003 and 2006 showed the lowest levels relative to the source populations, with 17 out of 18 comparisons for $H$ and 9 out of 18 comparisons for $A_r$ being significantly lower than the source populations (Wilcoxon rank sum tests, $p < 0.05$).

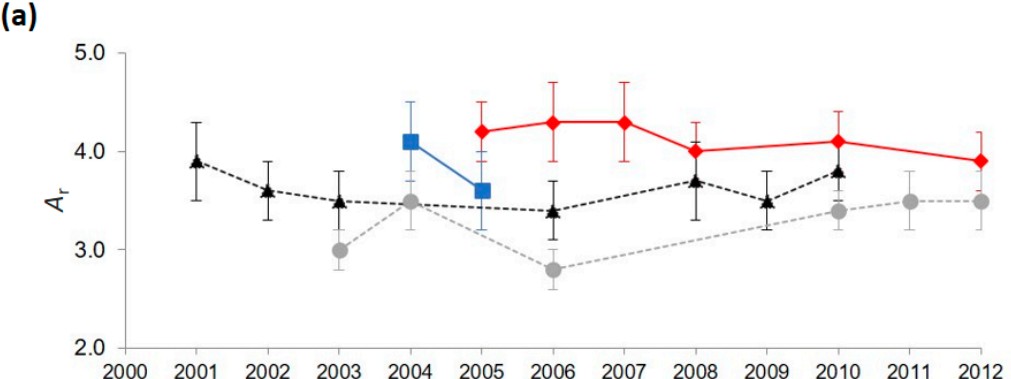

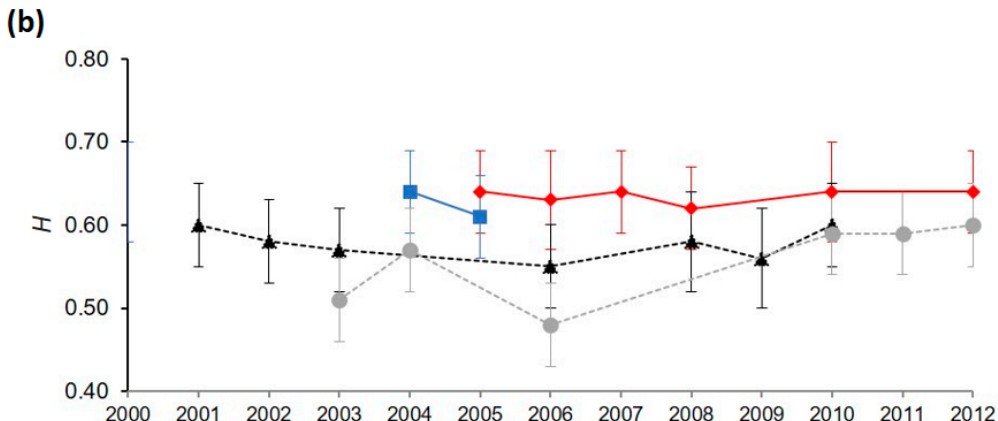

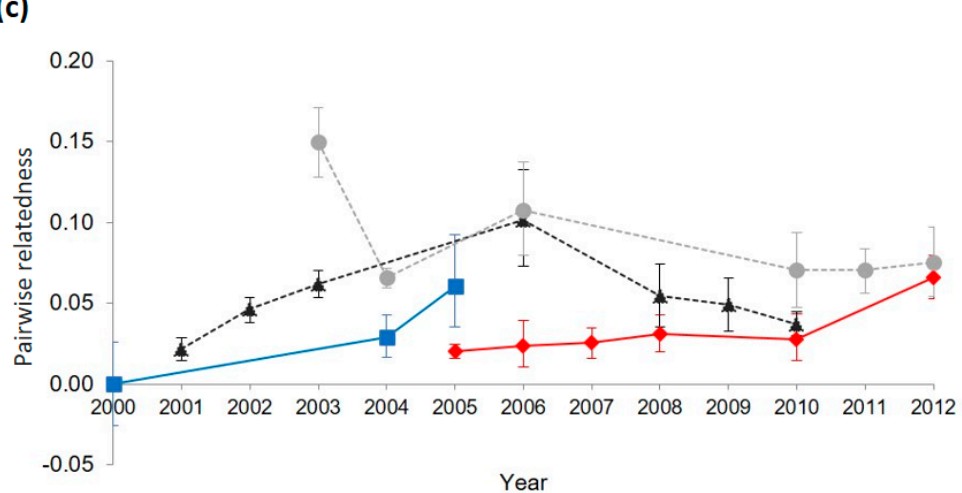

**Figure 2.** Estimates of (**a**) allelic richness ($A_r$); (**b**) gene diversity ($H$); and (**c**) pairwise relatedness within eastern (red) and western (blue) source, captive (black) and reintroduced Peniup Nature Reserve (grey) *Parantechinus apicalis* populations. Standard error bars are given around the means. Error bars around pairwise relatedness are bootstrapped 95% confidence limits.

**Table 3.** Estimates of genetic variation over 20 microsatellite loci within the source, captive and reintroduced *Parantechinus apicalis* populations. $N$ is an average sample size per locus. $A$ is the total number of alleles. $A_u$ is an average of unique alleles. $A_{(rare)}$ is an average number of rare alleles (frequency < 5%). $A_r$ is allelic richness. $H$ is gene diversity. $F_{IS}$ is inbreeding coefficient. $GD$ is genotypic disequilibrium. $N_e$ is an effective population size. Standard errors are given after mean values. Asterisks represent $F_{IS}$ values significantly different to zero at $p < 0.05$.

| Population | $N$ | $A$ | $A_u$ | $A_{(rare)}$ | $A_r$ | $H$ | $F_{IS}$ | $GD$ | $N_e$ | $N_e$ Range | Bottleneck |
|---|---|---|---|---|---|---|---|---|---|---|---|
| East | | | | | | | | | | | |
| 2005 | 40.2 ± 2.3 | 113 | 0.3 ± 0.2 | 1.2 ± 0.3 | 4.2 ± 0.3 | 0.64 ± 0.05 | 0.11 * | 0 | NA | NA | N |
| 2006 | 16.3 ± 0.3 | 103 | 0.1 ± 0.1 | 0.9 ± 0.4 | 4.3 ± 0.4 | 0.63 ± 0.06 | −0.02 | 0 | 15.0 | 11.1–21.4 | N |
| 2007 | 20.4 ± 0.3 | 108 | 0.1 ± 0.1 | 1.3 ± 0.4 | 4.3 ± 0.4 | 0.64 ± 0.05 | 0.02 | 0 | 42.9 | 26.9–90.3 | N |
| 2008 | 17.3 ± 0.2 | 93 | 0 | 0.5 ± 0.2 | 4.0 ± 0.3 | 0.62 ± 0.05 | 0.04 | 0 | 15.2 | 11.4–21.2 | Y |
| 2010 | 15.5 ± 0.5 | 96 | 0 | 1.0 ± 0.2 | 4.1 ± 0.3 | 0.64 ± 0.06 | 0.00 | 0 | NA | NA | N |
| 2012 | 19.0 ± 0.1 | 95 | 0 | 0.6 ± 0.2 | 3.9 ± 0.3 | 0.64 ± 0.05 | −0.04 | 0 | 9.4 | 7.5–11.6 | N |
| Overall | 141.3 ± 3.3 | 133 | 0.9 ± 0.3 | 2.4 ± 2.8 | 4.2 ± 0.4 | 0.64 ± 0.05 | 0.05 * | 1 | 74.1 | 52.5–110.9 | N |
| West | | | | | | | | | | | |
| 2000 | 6.7 ± 0.5 | 83 | 0.1 ± 0.1 | 0.1 ± 0.1 | NA | 0.64 ± 0.06 | 0.13 | 0 | NA | NA | - |
| 2004 | 16.8 ± 0.2 | 97 | 0.1 ± 0.1 | 0.6 ± 0.3 | 4.1 ± 0.4 | 0.64 ± 0.05 | 0.00 | 0 | 100.2 | 35.6–∞ | N |
| 2005 | 9.9 ± 0.1 | 79 | 0 | 0 | 3.6 ± 0.4 | 0.61 ± 0.05 | 0.03 | 0 | 42.3 | 18.8–∞ | N |
| Overall | 42.9 ± 0.9 | 112 | 0.4 ± 0.2 | 1.5 ± 1.6 | 4.0 ± 0.4 | 0.63 ± 0.05 | 0.03 | 0 | 54.1 | 36.7–91.4 | N |
| Captive founders | 24.7 ± 0.3 | 114 | 0 | 1.6 ± 2.2 | 4.3 ± 0.4 | 0.64 ± 0.05 | 0.03 | 0 | 69.7 | 40.0–204.1 | N |
| Captive population | | | | | | | | | | | |
| 2001 | 41.0 ± 0.0 | 96 | 0 | 0.6 ± 0.3 | 3.9 ± 0.4 | 0.60 ± 0.05 | 0.01 | 14 | 5.6 | 4.0–7.0 | N |
| 2002 | 38.2 ± 0.3 | 90 | 0 | 0.7 ± 0.3 | 3.6 ± 0.3 | 0.58 ± 0.05 | −0.08 | 7 | 5.3 | 3.8–6.9 | N |
| 2003 | 35.8 ± 0.1 | 83 | 0 | 0.5 ± 0.2 | 3.5 ± 0.3 | 0.57 ± 0.05 | −0.06 | 7 | 6.0 | 3.9–8.0 | N |
| 2006 | 15.0 ± 0.0 | 78 | 0 | 0.5 ± 0.2 | 3.4 ± 0.3 | 0.55 ± 0.05 | −0.07 | 0 | 2.1 | 1.9–2.5 | Y |
| 2008 | 18.9 ± 0.1 | 84 | 0 | 0.2 ± 0.1 | 3.7 ± 0.4 | 0.58 ± 0.06 | −0.10 | 3 | 2.0 | 1.8–2.3 | Y |
| 2009 | 18.7 ± 0.4 | 80 | 0 | 0.3 ± 0.1 | 3.5 ± 0.3 | 0.56 ± 0.06 | −0.03 | 1 | 4.2 | 3.0–6.1 | Y |
| 2010 | 36.9 ± 0.1 | 97 | 0 | 0.8 ± 0.3 | 3.8 ± 0.3 | 0.60 ± 0.05 | −0.02 | 2 | 9.8 | 8.1–11.7 | N |
| Overall | 220.4 ± 0.6 | 113 | 0 | 1.6 ± 2.6 | 4.0 ± 0.3 | 0.61 ± 0.05 | −0.01 | 50 | 24.5 | 21.0–28.5 | N |
| Reintroduced population | | | | | | | | | | | |
| 2003 | 13.7 ± 0.1 | 67 | 0 | 0.4 ± 0.1 | 3.0 ± 0.2 | 0.51 ± 0.05 | 0.00 | 0 | 4.0 | 2.6–7.8 | N |
| 2004 | 41.8 ± 0.4 | 85 | 0 | 0.7 ± 0.2 | 3.5 ± 0.3 | 0.57 ± 0.05 | −0.05 | 0 | 10.8 | 8.7–13.3 | N |
| 2006 | 9.0 ± 0.3 | 58 | 0.1 ± 0.1 | 0.1 ± 0.1 | 2.8 ± 0.2 | 0.48 ± 0.05 | −0.08 | 0 | 16.7 | 6.5–1890.3 | Y |
| 2010 | 12.5 ± 0.3 | 75 | 0 | 0.6 ± 0.2 | 3.4 ± 0.2 | 0.59 ± 0.05 | −0.03 | 0 | 5.4 | 3.0–8.8 | N |
| 2011 | 21.0 ± 0.0 | 83 | 0 | 0.7 ± 0.2 | 3.5 ± 0.3 | 0.59 ± 0.05 | −0.10 | 0 | 4.2 | 3.1–5.7 | N |
| 2012 | 15.0 ± 0.0 | 77 | 0 | 0.4 ± 0.1 | 3.5 ± 0.3 | 0.60 ± 0.05 | −0.03 | 0 | 8.2 | 5.6–11.8 | Y |
| Overall | 131.6 ± 1 | 103 | 0.1 ± 0.1 | 1.3 ± 1.2 | 3.7 ± 0.3 | 0.60 ± 0.05 | 0.01 | 13 | 16.7 | 14.5–19.1 | N |

A total of 155 alleles across 20 loci were detected. Of these, 38 (24.5%) were unique to the eastern source population and 17 (11.0%) were unique to the western source population. The eastern source population possessed a significantly higher number of unique alleles, on average, than any other population samples (Wilcoxon rank sum tests, $p < 0.01$ in all cases, Table 3). However, there were no significant differences in the average number of rare alleles between the population samples (Friedman rank sum test, $p = 0.858$). The wild-born individuals in the reintroduced population retained 13 (34.2%) and 6 (35.3%) of the unique alleles from the eastern and western source populations, respectively. However, they lost 9 (9.6%) to 30 (22.6%) alleles when compared to the source populations (Table 3). The largest loss was between the eastern source population and the founders (19 alleles, 14.3%). Only slight losses were observed between the founders and captive population (1 allele, 0.9%), and between the captive and reintroduced populations (10 alleles, 8.8%).

The estimates of $N_e$ were much lower in the captive population ($N_e = 24.5$, range 21.0 to 28.5) than the source populations (eastern source population, 74.1, range 52.5 to 110.9; western source population, 54.1, range 36.7 to 91.4, Table 3). The $N_e$ of the reintroduced population was comparable to the captive population, with an overall estimate of 16.7 (range 14.5 to 19.1). Population bottlenecks were also detected more frequently in the captive and reintroduced populations than the source populations (Table 3). All bottlenecks in these populations were detected after they had become established.

### 3.2. Population Structure of Captive and Reintroduced Populations

There was low, but significant, genetic differentiation between the source populations ($F_{ST}$ = 0.046). No significant temporal variation in $F_{ST}$ was detected within each of the source populations, but there was significant temporal variation in both the captive and reintroduced populations (Figure 3). Initially, the pairwise $F_{ST}$ values were lower between the captive and western source population than between the captive and eastern source population, but this changed with the opposite pattern evident in 2008–2010 (Figure 3). A similar pattern was observed in the reintroduced population, but mostly in the collection years 2010–2012 (Figure 3). Consistent with the pairwise $F_{ST}$ values, there were two genetic clusters detected in the captive population by the DAPC analysis separating the collection years 2000–2003 from 2006–2010 (Figure 4b). A similar pattern was observed in the reintroduced population for the collection years 2002–2007 and 2009–2012 (Figure 4c). This change coincided with the additional introductions of wild-caught individuals from the eastern source population to the captive population in 2007 and 2009, which subsequently increased the eastern ancestry proportion within the captive population (Figures 4a and 5). The change was not detected in the reintroduced population until 2009 (Figure 4c).

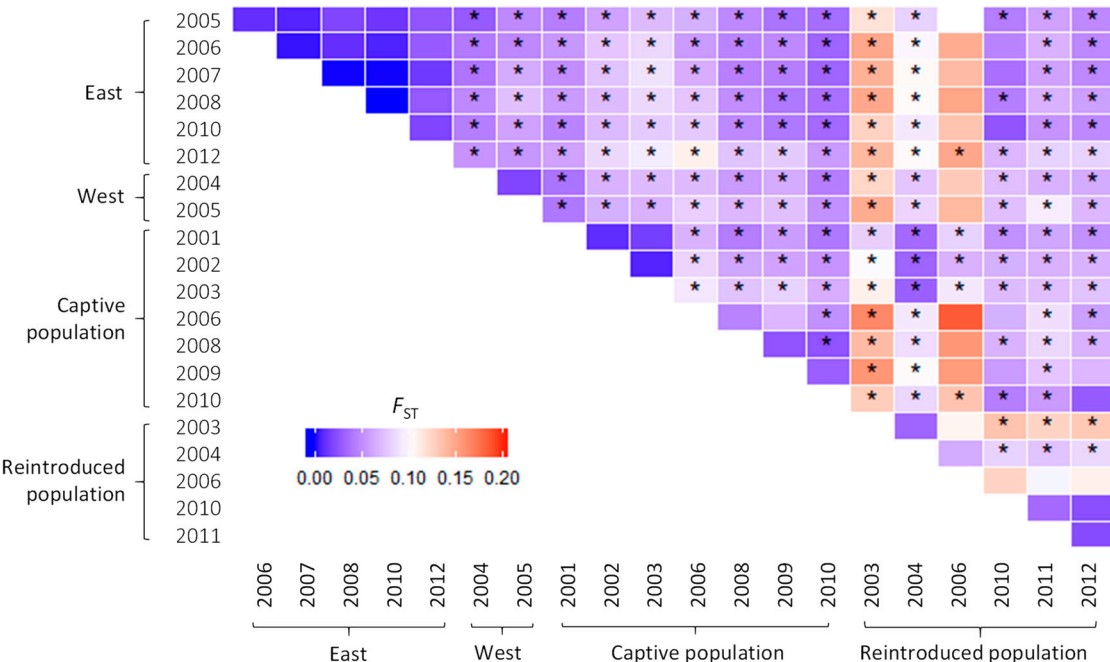

**Figure 3.** Pairwise $F_{ST}$ heatmap with $F_{ST}$ values ranging from 0 to 0.2 between source (east and west), captive and reintroduced populations of *Parantechinus apicalis*. $F_{ST}$ estimates significantly greater than zero ($p < 0.05$) after correction for multiple comparisons are marked with asterisks.

**(a) Source populations and founding individuals by years of capture**

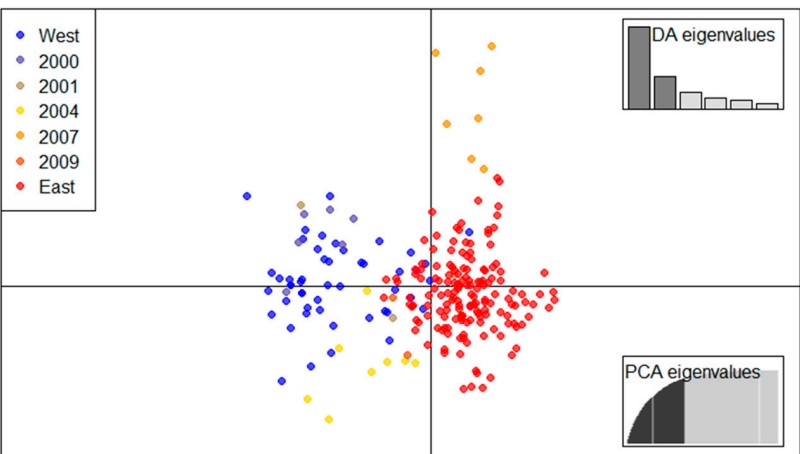

**(b) Captive population**

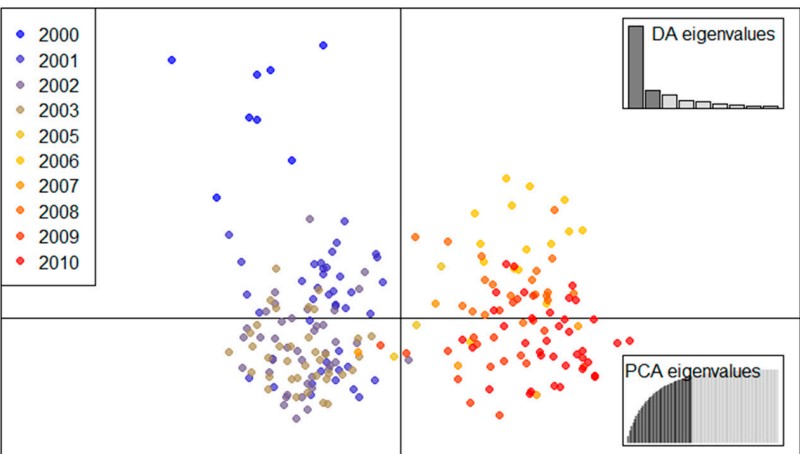

**(c) Reintroduced population**

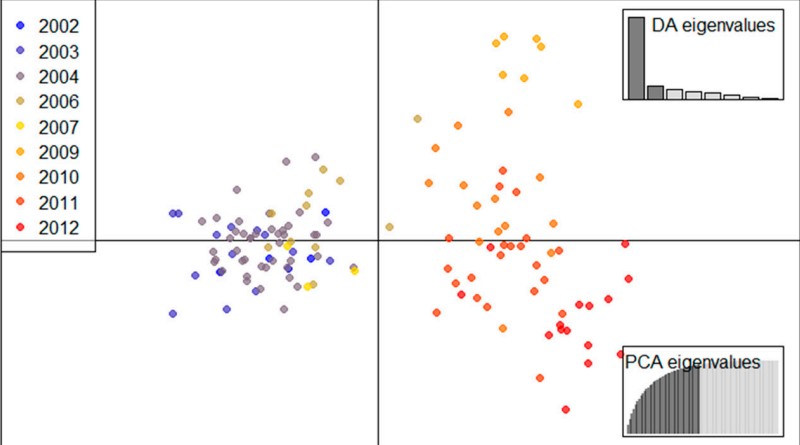

**Figure 4.** Scatterplot of the DAPC analysis showing the first two principal components. Clusters in different colors represent different collection years, except the source populations, which represented individuals pooled across the 2000–2012 collections. Dots represent *Parantechinus apicalis* individuals. Insets show the histogram of discriminant analysis eigenvalues. (**a**) Shows the source populations and founders introduced to the breeding program in different years; (**b**) shows the captive population; and (**c**) shows the reintroduced population.

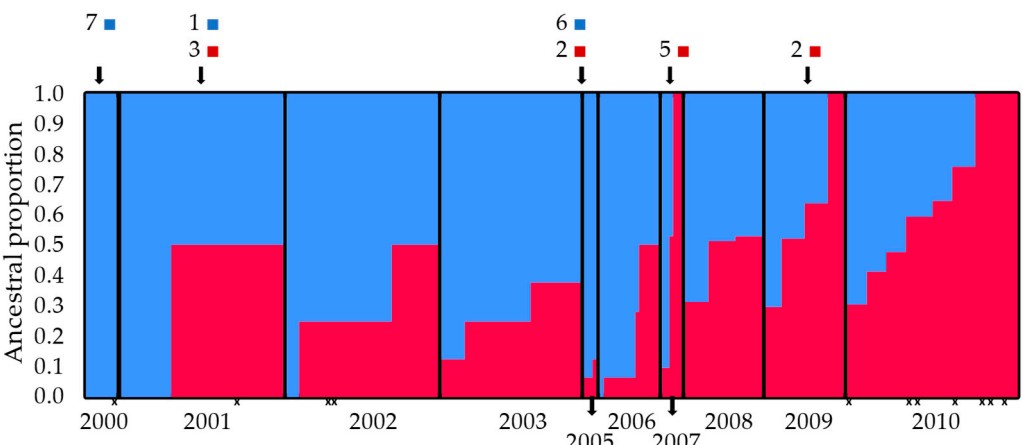

**Figure 5.** Ancestral proportion (blue: western subpopulation and red: eastern subpopulation) of captive-born *Parantechinus apicalis* calculated from the pedigree. Each bar represents one individual and black bars separate birth years. X marks individuals who died in captivity or were kept for next breeding season. Top inset indicates numbers of founders from different sources that contributed to the gene pool of the captive population at different time points. Note that the numbers do not match Table 1 because some founders did not contribute to the gene pool.

The multilocus $F_{IS}$ values of the captive- and Peniup-born dibblers were mostly negative, indicating a heterozygosity excess, but they were not significantly different from zero (Table 3). Significantly positive multilocus $F_{IS}$ values were observed in the eastern source population only (randomization tests, $p < 0.002$). The number of pairs of loci in genotypic disequilibrium (*GD*) ranged from 0 to 14. The highest number occurred in the captive population, especially during the first few generations, but it declined over time. The number of pairs of loci in *GD* in the source populations and reintroduced population ranged from zero to one (Table 3).

*3.3. Genetic Relatedness Comparisons*

The pairwise relatedness values of wild-born dibblers at the reintroduction site were consistently higher than wild-born dibblers in the source populations (Figure 2c, Wilcoxon rank sum tests, $p < 0.05$ in 47 out of 54 comparisons). The pairwise relatedness of the captive population showed a similar pattern and the values were comparable to the reintroduced population, except for 2003 (Figure 2c, Wilcoxon rank sum tests, $p < 0.05$). Overall, the pairwise relatedness was at the lowest in the founder group, then increased in the captive-born dibblers and increased again in the wild-born dibblers at the reintroduction site (Figure 6, Wilcoxon rank sum tests, $p < 0.01$ in all cases).

In the source populations, the pairwise relatedness values between pairs of females were significantly higher than male–male or female–male pairs (Wilcoxon rank sum tests, $p < 0.01$ in all comparisons, Figure 6). However, in the captive and reintroduced populations, the relatedness values of male–male pairs were similar or higher than female–female pairs (Wilcoxon rank sum tests, $p < 0.001$).

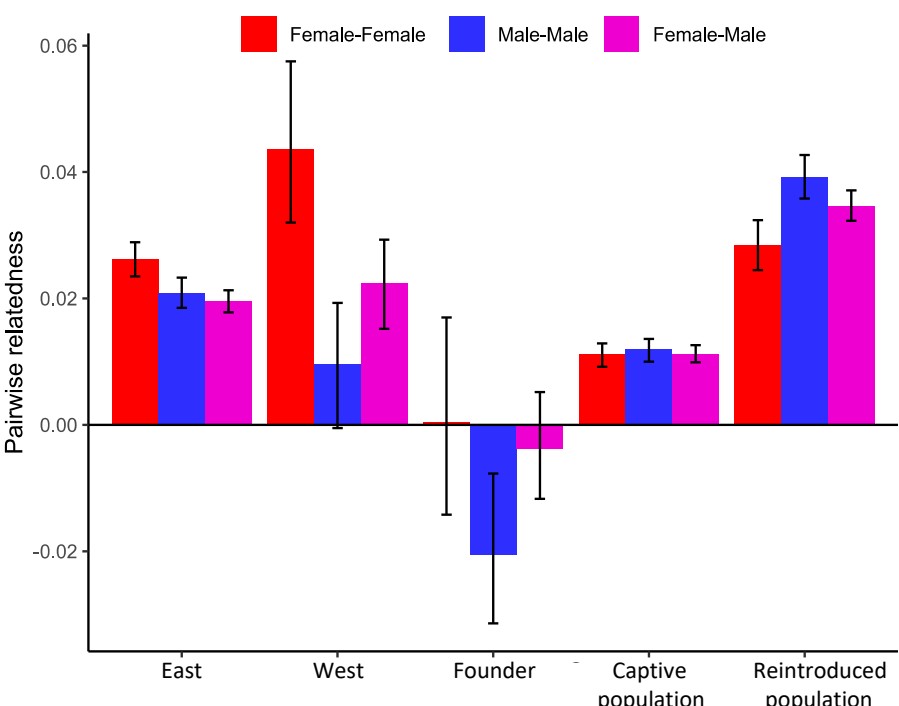

**Figure 6.** Mean pairwise genetic relatedness between female–female, male–male, and female–male *Parantechinus apicalis* pairs of the source populations (east and west), the wild-caught founders of the captive population, and the captive and reintroduced populations. Error bars are bootstrapped 95% confidence limits.

## 4. Discussion

### 4.1. Genetic Consequences of Mixing Subpopulations

Translocated populations often experience a significant loss of genetic variation and become genetically distinct from their source populations as a result of founder effects, genetic bottlenecks and/or genetic drift [12–14]. In this study, we show that the initial reintroduction resulted in a 10–16-fold reduction in $N_e$ compared to the source population, but then no significant further loss of genetic diversity was detected after 10 generations. This initial decline is similarly observed in other translocated populations [8,11,68]. In addition, several rare alleles were lost from the reintroduced population. This is not unexpected given the reduction in $N_e$, and rare alleles being more prone to loss following founder events and genetic bottlenecks than common alleles [69–71].

The maintenance of genetic diversity in the reintroduced population may be attributed to several factors. First, the number of founding animals (*n* = 26) used to establish the population was large enough to capture sufficient heterozygosity of the source populations, and the pairing strategy used by the captive breeding program was efficient to prevent significant losses. Generally, 30–50 individuals are recommended to capture 95% of heterozygosity or rare alleles with frequency less than 5% of the source population(s) [27,28]. A rapid population growth may also have shortened the amount of time the newly established population might have experienced severe genetic drift. This was supported by the lack of genetic bottleneck signatures detected during the early phase of reintroduction, and a steady increase in the effective population size in the reintroduced population after it became established until 2006 when the population crashed due to a lapse of effective predator control (A. Friend, pers. comm.). Similarly, a rapid population expansion was identified as the main factor for the high retention of genetic diversity in the white-tailed deer (*Odocoileus virginianus*) and European rabbit (*Oryctolagus cuniculus*) populations following their introductions [72,73]. Second, admixture from multiple source populations is likely to have bolstered the genetic variation in the reintroduced population, counteracting

subsequent losses [9,21–23]. In introduced populations of the brown anole (*Anolis sagrei*), a reduction in the genetic diversity, following a founder event, and an increase in genetic variation, due to admixture, were suggested to occur simultaneously, resulting in the maintenance of haplotype diversity in one population and higher haplotype diversity in another [74]. Finally, multiple releases of captive-bred individuals may have replenished the genetic diversity lost due to post-release mortality and variance in reproductive success amongst the founders [11,75]. Continuing releases of captive animals to the reintroduced population are also likely to have offset the genetic impacts of the population crash that occurred in 2006.

### 4.2. Consequences of Admixture on Population Structure

The captive and reintroduced populations in this study were established using individuals from two distinct genetic clusters within the Fitzgerald River NP [45]. The pedigree record of the captive population provided evidence of interbreeding between individuals from different genetic clusters. Based on the $F_{ST}$ values and DAPC analysis, both the captive and reintroduced populations were initially genetically more similar to the western subpopulation due to a higher proportion of founding animals from the western source population. After more individuals from the eastern subpopulation were introduced to the captive colony in 2007 and 2009, both the captive and reintroduced populations became genetically more similar to the eastern subpopulation. This highlights the relative importance of the different origins of the source populations in translocations and/or captive breeding programs, which can be used to manipulate the genetic ancestries within the translocated or captive population. For example, a manipulation of the Mexican wolf (*Canis lupus baileyi*) founders with three different ancestries was used to reduce the levels of inbreeding within a reintroduced population [76]. However, careful manipulations of this type are vulnerable to initial mortality and/or differential reproductive success among founders, as the result of local selection at the release site or mating preference [24,77].

### 4.3. Genetic Mixing and Relatedness

We found that the interbreeding of founders from different genetic clusters reduced the genetic relatedness among their progenies. This is not surprising given that dibblers from different genetic clusters were less likely to share alleles that are identical by descent. A similar finding was reported in farmed pearl oysters (*Pinctada margaritifera*). By pooling individuals from genetically divergent populations, it lowered the levels of pairwise relatedness when compared to the wild populations [78]. However, the reduction was short-lived due to limited mate availability and continued interbreeding within the new population. The pairwise relatedness of the female pairs was higher than the male pairs in both source populations, which reflects the male-biased dispersal pattern of dibblers [45]. A change in the dispersal behavior of males at the reintroduction site may have occurred as the genetic relatedness values between the pairs of males were much higher than in either of the source populations. This finding demonstrates that the initial genetic similarity between the founding individuals is important for the captive breeding and translocation programs, and high background inbreeding of founders can lead to higher levels of genetic relatedness and inbreeding among the offspring [20,79]. Increased levels of inbreeding can subsequently lead to failed translocations as a result of inbreeding depression, where the fitness of individuals is reduced from the expression of deleterious recessive alleles or genetic load [15,18,20]. To avoid this, obtaining founding animals from various locations, but from a similar habitat, can reduce the risk of selecting related individuals while also selecting founders and their offspring that can adapt to the release site [80].

### 4.4. Conservation Applications

This study shows that a large number of founders and rapid population growth can reduce gene diversity loss and maintain allelic richness in the reintroduced populations. If a large founder number cannot be achieved, multiple releases can counteract genetic loss

from mortality and variance in reproductive success among founders [11,75]. Furthermore, our study demonstrated that the genetic composition of the captive and reintroduced populations was influenced by the timing of founders from different subpopulations introduced to the captive breeding program. Although this was unintentional, it has a significant conservation implication, especially when mixing populations to preserve particular traits. Mixing between local and non-local populations needs careful management to avoid genetic swamping by one population or the other, as it can lead to maladaptation or the loss of desirable traits (e.g., [80]). Despite no significant loss of genetic diversity observed in this study, the reintroduced population still experienced a significant reduction in the effective population size relative to the wild source populations. The higher level of relatedness in the reintroduced population compared to the sources is a concern for long-term persistence as the population is small and isolated, thus is expected to experience a larger effect of genetic drift and continue to lose genetic diversity at a rate of $1/2\,N_e$ per generation [81]. Since 2012, the population has received an additional 69 captive animals in 2017 and continues to persist from the last camera trap monitoring in 2019. Continuous predator control is essential for both long- and short-term persistence, as shown in 2006 when the population crashed from predation. Habitat corridors, known as Gondwana Link, have initiated in 2007 to reconnect the Fitzgerald River National Park and the Stirling Ranges, and Peniup Nature Reserve is one of the important steppingstones [82]. This corridor is crucial for the long-term persistence of this species to assist in migration and to expand its population size. While the corridor is still under restoration, interval top-ups (<20% of the recipient population size) of animals from one or both sources are recommended to facilitate gene flow into the population. With the advancement in genomic technology, a follow-up genome-wide study would provide further insight to gain a better understanding of the diversity and admixture in these populations.

**Supplementary Materials:** The following are available online at https://www.mdpi.com/article/10.3390/d13060257/s1, Table S1: characteristics of the 21 microsatellite loci that were selected for use in characterizing the genetic variability of the dibbler, *Parantechinus apicalis*.

**Author Contributions:** Conceptualization, R.T., W.J.K., and H.R.M.; methodology, R.T., W.J.K., and H.R.M.; software, R.T.; validation, W.J.K., H.R.M., and K.O.; formal analysis, R.T.; investigation, R.T.; resources, J.A.F.; data curation, R.T.; writing—original draft preparation, R.T.; writing—review and editing, W.J.K., H.R.M., J.A.F., and K.O.; visualization, R.T.; supervision, W.J.K. and H.R.M.; project administration, R.T.; funding acquisition, H.R.M. All authors have read and agreed to the published version of the manuscript.

**Funding:** This research was funded by the Department of Biodiversity, Conservation and Attractions (DBCA) and the University of Western Australia (UWA).

**Institutional Review Board Statement:** This study was approved by UWA Animal Ethics Committee (AEC: 16A/2012), the Zoological Parks Authority Animal Ethics Committee (SOP 49063 and 24252), and DBCA (DEC AEC: 66/2009).

**Informed Consent Statement:** Not applicable.

**Data Availability Statement:** Raw microsatellite genotypes are available via Mendeley Data doi:10.17632/mrpjrm5kn8.1.

**Acknowledgments:** We are grateful to Perth Zoo and DBCA for supplying tissues and associated data. We thank Cathy Lambert for supplying tissues from the captive breeding program, Yvette Hitchen for laboratory support and Tim Button for technical support.

**Conflicts of Interest:** The authors declare no conflict of interest.

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
