# Peer review of "Temporal Variation in the Genetic Composition of an Endangered Marsupial Reflects Reintroduction History"

_diversity, doi:10.3390/d13060257_

Round 1

Reviewer 1 Report

Thavornkanlapachai et al provide a comprehensive study of the genetic consequences of a captive breeding and translocation program for the dibbler. This was an interesting read and provides an important analysis for the conservation of this species. Their methods are sound and clearly described. However, the discussion lacks some inspiration and I was hoping to see more discussion about what is next for this species based on these results, how can their findings inform the next steps in conserving this species and what should be done to improve the success of the conservation program? Following improvements to the discussion, this article is suitable for publication in Diversity.

More specific comments:

Discussion

Lines 288-291. These two conclusions read as if they contradict each other, but I think it is just the way it is phrased. Perhaps say that the initial translocation resulted in a 10-16 fold reduction in Ne compared to the source population, but then no significant further loss of genetic diversity was detected after 10 generations.

Lines 296-301: So, are you saying that only 26 founding animals was enough to capture the majority of alleles in the source population as the source population was genetically poor? And this does seem to contradict the 10-16 fold decrease in Ne that you report – if Ne reduced so much then surely a lot of genetic variation was not present in the translocated population that was found in the source populations? I think the second point in this section (admixture) is more likely an explanation.

Lines 329-331: This is really stating the obvious. Perhaps something more insightful here would be to consider the relative importance of the different origins of the source populations, as highlighted by the Mexican wolf example you mention.

Lines 333-335: Do you see evidence for differential mortality and/or reproductive success among founders?

Section 4.3: What might be the consequences for the population of the heightened relatedness and therefore greater inbreeding in the translocated population? How could these consequences be avoided/reduced?

Line 355: do you mean ‘counteract’ rather than ‘encounter’?

Lines 356-360: I’m not sure what you mean here – the lower the differentiation between populations, the less the increase in genetic diversity when mixing them? Again, this is an obvious point. And “lowered the genetic similarity between admixed individuals in the early generations” compared to what? Genetic similarity in the source populations? This isn’t correct as you have higher Ne in the source populations. This section needs clarifying.

Line 360-361: Again, this is not a novel conclusion to come to. Can you take it further to make specific suggestions for the dibbler: where is the centre of diversity? Which populations should be sourced from to continue to improve the viability of the translocated population? What’s next for this population, now you have demonstrated that relatedness is much higher in this population and there is a great risk of inbreeding depression?

Line 366-368: Would this really be a concern for such an endangered species distributed over a very small area though? You’d be better off trying to increase the numbers as much as you can and increasing diversity as much as you can so that there is a reduced effect of genetic drift and an increased diversity for selection to act on.

Line 368: In this study you only look at a set of microsatellite markers. I am left wondering what the genetic diversity losses might look like if you consider the wider genome. Perhaps suggesting a follow-up genome wide approach could be useful for gaining a greater understanding of the diversity and admixture in these populations.

Grammatical:

Line 54: ‘P. apicalis’ – write out in full at start of sentence

Line 300: Fewer animals, not less

Author Response

Reviewer one

Comments and Suggestions for Authors

Thavornkanlapachai et al provide a comprehensive study of the genetic consequences of a captive breeding and translocation program for the dibbler. This was an interesting read and provides an important analysis for the conservation of this species. Their methods are sound and clearly described. However, the discussion lacks some inspiration and I was hoping to see more discussion about what is next for this species based on these results, how can their findings inform the next steps in conserving this species and what should be done to improve the success of the conservation program? Following improvements to the discussion, this article is suitable for publication in Diversity.

More specific comments:

Discussion

Lines 288-291. These two conclusions read as if they contradict each other, but I think it is just the way it is phrased. Perhaps say that the initial translocation resulted in a 10-16 fold reduction in Ne compared to the source population, but then no significant further loss of genetic diversity was detected after 10 generations.

Response – Rephrased the sentence as suggested.

Lines 296-301: So, are you saying that only 26 founding animals was enough to capture the majority of alleles in the source population as the source population was genetically poor? And this does seem to contradict the 10-16 fold decrease in Ne that you report – if Ne reduced so much then surely a lot of genetic variation was not present in the translocated population that was found in the source populations? I think the second point in this section (admixture) is more likely an explanation.

Response – We have rephrased that as the gene diversity not allele diversity (line 346). As showed in the result, the animals born at the translocation site only retained 34.2% and 35.3% of unique alleles from eastern and western sources respectively. Nevertheless, heterozygosity is less sensitive to loss of genetic diversity as it reduces at a rate of ½Ne per generation while allele diversity is more sensitive to losses. Admixture would also counter any initial losses.

Lines 329-331: This is really stating the obvious. Perhaps something more insightful here would be to consider the relative importance of the different origins of the source populations, as highlighted by the Mexican wolf example you mention.

Response – We have rewritten the sentence in this section as suggested (line 379-385).

Lines 333-335: Do you see evidence for differential mortality and/or reproductive success among founders?

Response – When we released the captive bred animals, individuals had mixed ancestries due to the captive breeding design. There was no obvious differential mortality and/or reproductive success in the released animals to skew the admixture proportions. Founders from the source populations underwent mate pair selection using a pedigree program called Spark which pairs the animals based on their mean kinship. There was no obvious differential mortality and/or reproductive success among founders in captivity as well. However, in other wild to wild translocation studies, they have reported high mortality rate as much as half of the released animals or mating preference based on males’ body size. We changed the word ‘differential’ to ‘initial’ to indicate morality in the initial phase (line 384).

Section 4.3: What might be the consequences for the population of the heightened relatedness and therefore greater inbreeding in the translocated population? How could these consequences be avoided/reduced?

Response - Increase levels of inbreeding can subsequently lead to fail translocation as a result of inbreeding depression where fitness of individuals is reduced from expression of deleterious recessive alleles or genetic load [14,17,19]. To avoid this, selecting founding animals from various locations, but from a similar habitat, can reduce the risk of selecting related individuals as well as selecting founders and their offspring that can adapt to the release site [76]. This additional explanation has been added to section 4.3 (line 401-406).

Line 355: do you mean ‘counteract’ rather than ‘encounter’?

Response – This has been corrected.

Lines 356-360: I’m not sure what you mean here – the lower the differentiation between populations, the less the increase in genetic diversity when mixing them? Again, this is an obvious point. And “lowered the genetic similarity between admixed individuals in the early generations” compared to what? Genetic similarity in the source populations? This isn’t correct as you have higher Ne in the source populations. This section needs clarifying.

Response – This paper is one of three parts of a larger study which looks at the admixture consequences from three different levels of divergence of the source populations (subpopulations – this study, short-term isolated island populations, and long-term isolated island populations). We found that the increased level of heterozygosity in the translocated population was positively correlated to the divergence level between the source populations. Since this section is unclear without this additional content, we have removed it instead of adding it as it can take away the main message.

Line 360-361: Again, this is not a novel conclusion to come to. Can you take it further to make specific suggestions for the dibbler: where is the centre of diversity? Which populations should be sourced from to continue to improve the viability of the translocated population? What’s next for this population, now you have demonstrated that relatedness is much higher in this population and there is a great risk of inbreeding depression?

Response – The dibbler persists only in a single wild population on mainland Australia, with genetic structuring present within this population. Both East and West subpopulations contain similar levels of genetic diversity. They also shared a large number of alleles as well as contain some unique alleles so recommending to sample from various locations was the only suggestion that we can be made for sourcing future founding animals. The higher level of relatedness in the translocated population compared to the sources can be a concern for long-term persistence as the population is small and isolated and expected to experience larger effect of genetic drift and continue to lose genetic diversity at a rate of 1/2Ne per generation. To support long-term persistence, a habitat corridor to the source populations, continue predator control and interval top-ups of animals from one or both sources are recommended to increase population size and facilitate gene flow into the population (line 420-435).

Line 366-368: Would this really be a concern for such an endangered species distributed over a very small area though? You’d be better off trying to increase the numbers as much as you can and increasing diversity as much as you can so that there is a reduced effect of genetic drift and an increased diversity for selection to act on.

Response – We have included the comments on increase population size and habitat corridor (line 427-431).

Line 368: In this study you only look at a set of microsatellite markers. I am left wondering what the genetic diversity losses might look like if you consider the wider genome. Perhaps suggesting a follow-up genome wide approach could be useful for gaining a greater understanding of the diversity and admixture in these populations.

Response – This recommendation on future direction has been included (line 433-435).

Grammatical:

Line 54: ‘P. apicalis’ – write out in full at start of sentence

Response – Done.

Line 300: Fewer animals, not less

Response – Done.

Reviewer 2 Report

This is an important and well executed study into the genetic management implications of captive breeding and reintroduction.  It very thoroughly examines different aspects of how the genetic constituents of founder source animals can be managed and the impacts of these differing strategies on long term genetic diversity of the reintroduced population compared with the founder populations over time.  It is a very significant contribution to conservation biology of wider relevance than to the threatened species studied.

Author Response

Thank you for reviewing our paper and your kind comments. 

Reviewer 3 Report

Review of diversity-1224235

“Temporal variation in the genetic composition of a newly established population of a small marsupial, the dibber, reflects translocation history”

Dear Authors,

This is an interesting and important case study of the impact of one translocation strategy on genetic composition of the new population. I enjoyed learning about your findings. Thank you for doing this work.

The study is well designed and documented. The manuscript is well written and the figures and tables useful and not excessive. I have only minor suggestions for improvement. Substantive comments are below, while those of a more editorial nature follow.

Substantive Comments:

1) The Introduction does not do the work justice. It was not until I was well into the Methods and Results that I really began to grasp what, specifically, the authors were testing. This is a shame. I suggest substantially bolster the Introduction on what specific questions the authors were testing and why. Some of this is clearly in the Discussion and Conclusions now and should be first mentioned in the Introduction, ideally as hypotheses or predictions.

2) It would be helpful for readers with no knowledge of conservation efforts for this species if a new table was generated that include the particulars of the 7 (I believe) releases of Peniup Nature Reserve. For each of the 7 releases, this should include total number (already in Table 1), as well as the sex, age-class, and source population (East or West from FRNP).

3) Some of the major conclusions (conservation implications) should be included at the end of the Abstract for those readers that don’t read the whole text.

Editorial Comments:

Line 2: The title is rather long and not terribly catchy to attract readers. I suggest significantly shortening it. Perhaps consider something along the lines of: “Temporal variation in the genetic composition of an endangered marsupial reflects reintroduction history” or “Genetic assessment of a reintroduced population of an endangered marsupial”

Line 35: Delete “and introduced species”. This is encompassed in human activities.

Line 36: Sorry, some semantics to consider: “translocation” can mean many things, including reintroducing species to their native range where they are locally extirpated, “augmenting” populations with new individuals to bolster a small population, and “introductions” of species to habitats outside their natural distributional range. Please be specific in the Introduction and Discussion that this is a reintroduced population of an endangered species, and that population augmentation occurred after the initial reintroduction.

Line 38: Perhaps replace “continuity” with “persistence”?

Line 76: How many dibblers were involved in the initial reintroduction event in 2001?

Line 78: How many dibblers were released to augment the reintroduced population?

Line 90: There is no description at all of the study site. Either provide a description or remove it from the subtitle.

Line 91-96: These sentences appear out of place. Please consider moving them to line 107 – before “A total of 156 samples…”

Line 111: Small point, but please indicate how the tissue sample was taken. If a biopsy punch then indicate the size of the punch and any other relevant details.

Line 112: DMSO = spell out at first use.

Line 115: In this figure caption and, I believe, elsewhere in there is reference to eastern and western “lineages”. These are not evolutionary lineages in the sense I would use that word. Please consider change to “population” or “subpopulation”, as appropriate.

Line 115, 119 and XXX: Figure captions are often not “stand alone”. For example, the species is not included. Please provide common name and scientific name for species in the figures.

Line 119: Here and elsewhere tables break across pages. This is inconvenient for readers. Please replace these so that the whole table is on the same page.

Line 126: Please place a space after each table the continuation of the text. As it stands now, it takes a moment to sort out if the text after the tables is a footnote or not.

Line 248: Spell out the genus here.

Lines 296-319: This paragraph is excellent. It would be great to include some of this in the Introduction to set the stage better for this work.

Line 312: Place the scientific name here in parentheses to be consistent with similar instances in the text.

Line 352: The concluding paragraph is well done. It is a shame that some of these main conclusions and recommendations are not highlighted in the Abstract, given that many readers will only read the Abstract.

Line 366: delete “a” before “careful”

Author Response

Reviewer three

Review of diversity-1224235

“Temporal variation in the genetic composition of a newly established population of a small marsupial, the dibber, reflects translocation history”

Dear Authors,

This is an interesting and important case study of the impact of one translocation strategy on genetic composition of the new population. I enjoyed learning about your findings. Thank you for doing this work.

The study is well designed and documented. The manuscript is well written and the figures and tables useful and not excessive. I have only minor suggestions for improvement. Substantive comments are below, while those of a more editorial nature follow.

Substantive Comments:

1) The Introduction does not do the work justice. It was not until I was well into the Methods and Results that I really began to grasp what, specifically, the authors were testing. This is a shame. I suggest substantially bolster the Introduction on what specific questions the authors were testing and why. Some of this is clearly in the Discussion and Conclusions now and should be first mentioned in the Introduction, ideally as hypotheses or predictions.

 Response – We added another paragraph which adds more background on genetic mixing, the recommended number of released animals, captive breeding program, and releasing strategies (line 57-75).

2) It would be helpful for readers with no knowledge of conservation efforts for this species if a new table was generated that include the particulars of the 7 (I believe) releases of Peniup Nature Reserve. For each of the 7 releases, this should include total number (already in Table 1), as well as the sex, age-class, and source population (East or West from FRNP).

 Response – The additional information about the released animals is included in Table 2. These released animals were either purebreds or admixed depending on when the new founders were introduced to the captive breeding program. A visual summary is added as Figure 5 in the result section.

3) Some of the major conclusions (conservation implications) should be included at the end of the Abstract for those readers that don’t read the whole text.

 Response – Conservation implications have been added to the abstract.

Editorial Comments:

Line 2: The title is rather long and not terribly catchy to attract readers. I suggest significantly shortening it. Perhaps consider something along the lines of: “Temporal variation in the genetic composition of an endangered marsupial reflects reintroduction history” or “Genetic assessment of a reintroduced population of an endangered marsupial”

 Response – Thank you for the title suggestion. We changed it to the first one.

Line 35: Delete “and introduced species”. This is encompassed in human activities.

 Response – Done.

Line 36: Sorry, some semantics to consider: “translocation” can mean many things, including reintroducing species to their native range where they are locally extirpated, “augmenting” populations with new individuals to bolster a small population, and “introductions” of species to habitats outside their natural distributional range. Please be specific in the Introduction and Discussion that this is a reintroduced population of an endangered species, and that population augmentation occurred after the initial reintroduction.

 Response – We have added the three different types of translocations to the introduction section (line 38-42) and changed the term “translocation” to “reintroduction” when refer to the dibbler reintroduction to the Peniup Nature Reserve but kept the term “translocation” when refer to all types of translocations in general.

Line 38: Perhaps replace “continuity” with “persistence”?

 Response – Done.

Line 76: How many dibblers were involved in the initial reintroduction event in 2001?

 Response – 41. This information is in Table 1 (under the Released column) and the added Table 2.

Line 78: How many dibblers were released to augment the reintroduced population?

 Response – This information is in Table 1 and the added Table 2.

Line 90: There is no description at all of the study site. Either provide a description or remove it from the subtitle.

 Response – We have removed it.

Line 91-96: These sentences appear out of place. Please consider moving them to line 107 – before “A total of 156 samples…”

 Response – The order of the sentences has been rearranged to aid the flow of the timeline. We also moved ethic approval declaration to the end of the paragraph (line 132-136).

Line 111: Small point, but please indicate how the tissue sample was taken. If a biopsy punch then indicate the size of the punch and any other relevant details.

 Response – Done.

Line 112: DMSO = spell out at first use.

 Response – Done.

Line 115: In this figure caption and, I believe, elsewhere in there is reference to eastern and western “lineages”. These are not evolutionary lineages in the sense I would use that word. Please consider change to “population” or “subpopulation”, as appropriate.

Response – Done.

 Line 115, 119 and XXX: Figure captions are often not “stand alone”. For example, the species is not included. Please provide common name and scientific name for species in the figures.

 Response – We added the scientific names to the figures and tables.

Line 119: Here and elsewhere tables break across pages. This is inconvenient for readers. Please replace these so that the whole table is on the same page.

 Response – Done.

Line 126: Please place a space after each table the continuation of the text. As it stands now, it takes a moment to sort out if the text after the tables is a footnote or not.

 Response – Done.

Line 248: Spell out the genus here.

 Response – Done.

Lines 296-319: This paragraph is excellent. It would be great to include some of this in the Introduction to set the stage better for this work.

 Response – This section has been added as part of the introduction (line 69-71).

Line 312: Place the scientific name here in parentheses to be consistent with similar instances in the text.

 Response – Done.

Line 352: The concluding paragraph is well done. It is a shame that some of these main conclusions and recommendations are not highlighted in the Abstract, given that many readers will only read the Abstract.

  Response – Done.

Line 366: delete “a” before “careful”

 Response – Done.